# Vitiligo and anxiety: A systematic review and meta-analysis

**Assiya Kussainova**[1], **Laura Kassym**[2]*, **Almira Akhmetova**[1], **Natalya Glushkova**[3‡], **Ulugbek Sabirov**[4], **Saltanat Adilgozhina**[5], **Raikhan Tuleutayeva**[6], **Yuliya Semenova**[7]

1 Department of Dermatovenerology and Cosmetology, NJSC "Semey Medical University", Semey, Republic of Kazakhstan, 2 Nazarbayev University School of Medicine, Nur-Sultan, Republic of Kazakhstan, 3 Department of Epidemiology, Evidence-Based Medicine and Biostatistics, Kazakhstan Medical University Higher School of Public Health, Almaty, Republic of Kazakhstan, 4 Republican Specialized Scientific and Practical Medical Center of Dermatovenerology and Cosmetology, Tashkent, Republic of Uzbekistan, 5 Department of Family Medicine, NJSC "Semey Medical University", Semey, Republic of Kazakhstan, 6 Department of Pharmacology Department, NJSC "Semey Medical University", Semey, Republic of Kazakhstan, 7 Department of Neurology, Ophthalmology, Otorhinolaryngology, NJSC "Semey Medical University", Semey, Republic of Kazakhstan

☯ These authors contributed equally to this work.
‡ These author also contributed equally to this work.
* laura.kassym@gmail.com

**Data Availability Statement:** All relevant data are within the manuscript and its Supporting Information files.

**Funding:** The author(s) received no specific funding for this work.

## Abstract

### Background

Vitiligo is an acquired depigmenting skin disease which is often accompanied by mental distress. There are numerous studies dedicated to local and global prevalence of depression in patients with vitiligo but anxiety has not been recognized as a major mental problem within named population. We aimed to evaluate the prevalence of anxiety among patients with vitiligo from different countries and to compare it with patients suffering from eczema, psoriasis, and acne.

### Methods

In November 2019, we conducted a systematic search for observational studies that examined the prevalence of anxiety in vitiligo patients. Fifteen studies comprising 1176 patients with vitiligo were included to our systematic review.

### Results

The general prevalence of anxiety among vitiligo patients was equal to 35.8%. Statistically significant difference in anxiety rates was found among female and male patients (47.32% vs 42.4%) ($P$ = 0.03), but the clinical relevance of this issue remains arguable. In addition, the pooled odds ratio among vitiligo and non-vitiligo patients did not indicate a statistical significance among patients coming from different continents.

### Conclusions

The pooled prevalence of anxiety among vitiligo patients worldwide was comparable to other severe skin disorders. This finding accentuates the necessity of anxiety awareness in management of patients with skin diseases.

**Competing interests:** The authors have declared that no competing interests exist.

## Introduction

Psychogenic effects related to various health disorders have become the issue of growing discussion in scientific literature over past decades [1, 2]. Globally, there is an increasing rate of anxiety disorders–a group of mental health problems characterized by the feelings of worry and uneasiness that are commonly generalized and present an overreaction to a problem that appears to be threatening [3]. Anxiety disorders quite seldom occur alone and are frequently associated with depression or other mental health problems [4].

The sample of such population groups could be made of vitiligo, which is an acquired lasting skin disorder. Although the etiology of vitiligo is not fully understood yet, common manifestation includes the patches of depigmentation with typically sharp margins [5]. Although the global rate of vitiligo is approximately 1%, some populations show twofold to threefold increase in rates [6]. These patches of skin depigmentation tend to expand with time and affected individuals experience a range of emotional problems. In certain cultures individuals suffering from vitiligo may be stigmatized and could experience difficulty with finding a couple or staying employed [7]. Inevitably, this worsens psychological distress and might even lead to a suicide attempt, especially if vitiligo affects visible body parts [8].

Although the relatively many papers devoted to the issue of anxiety in vitiligo patients had been published before, there is no pooled evidence that is needed for comprehensive understanding of this problem. A 2017 systematic review and meta-analysis on the prevalence of depression among patients with vitiligo estimated that the pooled prevalence of depression was 0.253 across 25 studies [9]. A 2018 meta-analysis of the prevalence and odds of depression in patients with vitiligo found a wide range of prevalence between 8% and 33% across 17 studies, depending on the diagnostic tool used. [10]. Meanwhile, anxiety in patients with vitiligo warrants higher awareness and greater attention as it can negatively affect adherence to treatment and overall quality of life [11]. As people with vitiligo appear to experience psychological problems with higher frequency than general population, the assessment of psychological state should be performed during routine clinical evaluation [12, 13].

Such, the existing data indicate that patients with vitiligo possibly face a higher risk of mental distress, although the current evidence coming from pooled analyses is insufficient. In this systematic review the null hypothesis was that there is no difference in prevalence of anxiety among vitiligo and non-vitiligo persons. Therefore, the aim of this study was to evaluate the prevalence of anxiety among patients with vitiligo from different countries and to compare it with patients suffering from eczema, psoriasis, and acne by conducting a systematic review and meta-analysis of published observational studies.

The specific goals of the present systematic review and meta-analysis are:

- To determine the prevalence of anxiety in vitiligo patients in comparison with non-vitiligo patients.

- To investigate the impact of some variables such as gender, continent, type of skin disorder on anxiety rate among vitiligo and non-vitiligo patients.

## Materials and methods

We conducted this systematic review and meta-analysis in accordance with the Preferred Reporting Items for Systematic Reviews and Meta-Analyses (PRISMA) statement [14]. Prior to quantitative and systematic synthesis we retrieved all studies that were targeted on assessment of associations between vitiligo and anxiety.

## Search strategy

A comprehensive database search was performed independently by two co-authors (A.K. and L.K.) using Pubmed, PsycINFO and Cochrane Library databases. An initial literature search in the mentioned databases used such keywords as "vitiligo" and "anxiety". The following search criteria were applied for PubMed (Ovid MEDLINE): ["Vitiligo"(MeSH)] AND ["Anxiety"(-MeSH) OR "Anxiety Disorder" (MeSH) OR Anx* (title/abstract;TIAB)] (see S1 Table). This search was limited to English-language studies published from inception to 30 November 2019. Also, we looked for publications in Russian language, for which reason we applied for Cyberleninka and eLIBRARY databases to screen for studies published from inception to 30 November 2019. Unfortunately, we failed to identify such publications despite the careful search and for this reason we only included studies published in English. Subsequently, we evaluated the abstracts of all identified papers to determine if they meet the inclusion criteria. Finally, we screened the reference lists of all eligible articles in order to find additional relevant articles.

## Inclusion criteria

Our inclusion criteria were as follows: (i) studies that included vitiligo patients; (ii) studies that assessed the prevalence of anxiety; (iii) studies that evaluated the prevalence of anxiety self-reporting or examination by a psychiatrist; (iv) studies that were published in English and (v) studies that were considered to be of high and medium-quality by Newcastle-Ottawa scale ($\geq$4 points).

## Exclusion criteria

Our exclusion criteria were as follows: (i) studies that did not state the rate of anxiety among vitiligo patients or provided insufficient data for calculation of anxiety rates; (ii) unavailability of the full text for full review; (iii) studies with low methodological quality, i.e. case reports, case series and commentaries; (iv) studies published in other languages apart from English and (v) studies that were considered to be of low-quality by Newcastle-Ottawa scale (<4 points). Besides, we did not include the studies that reported on anxiety during or after such psychotraumatic events as a war, natural or man-induced disaster, epidemic because they enable confounder effects.

## Article selection

The initial search and selection of articles was performed independently by two co-authors (A. K and L.K.), who screened for titles and abstracts and excluded all articles that did not meet the inclusion criteria. As a next step, we retrieved the full texts of articles that were considered to be eligible and evaluated all studies on the basis of their design. Any differences of opinion on study eligibility were resolved in discussions with Y.S. The selection process following PRISMA guidelines is presented in Fig 1.

The studies had to report sufficient data, such as odds ratio (OR) and 95% confidence interval (CI), so that the corresponding standard errors (SEs) could be calculated. If such information was not available, we looked for the crude data with the number of cases.

## Data extraction and study evaluation

Two reviewers (A.K. and L.K.) extracted the data from the selected articles. All selected data were arranged as a standardized form which contained:(i) last name of first author and year of publication;(ii) country of the study origin; (iii) the sample size of vitiligo and non-vitiligo

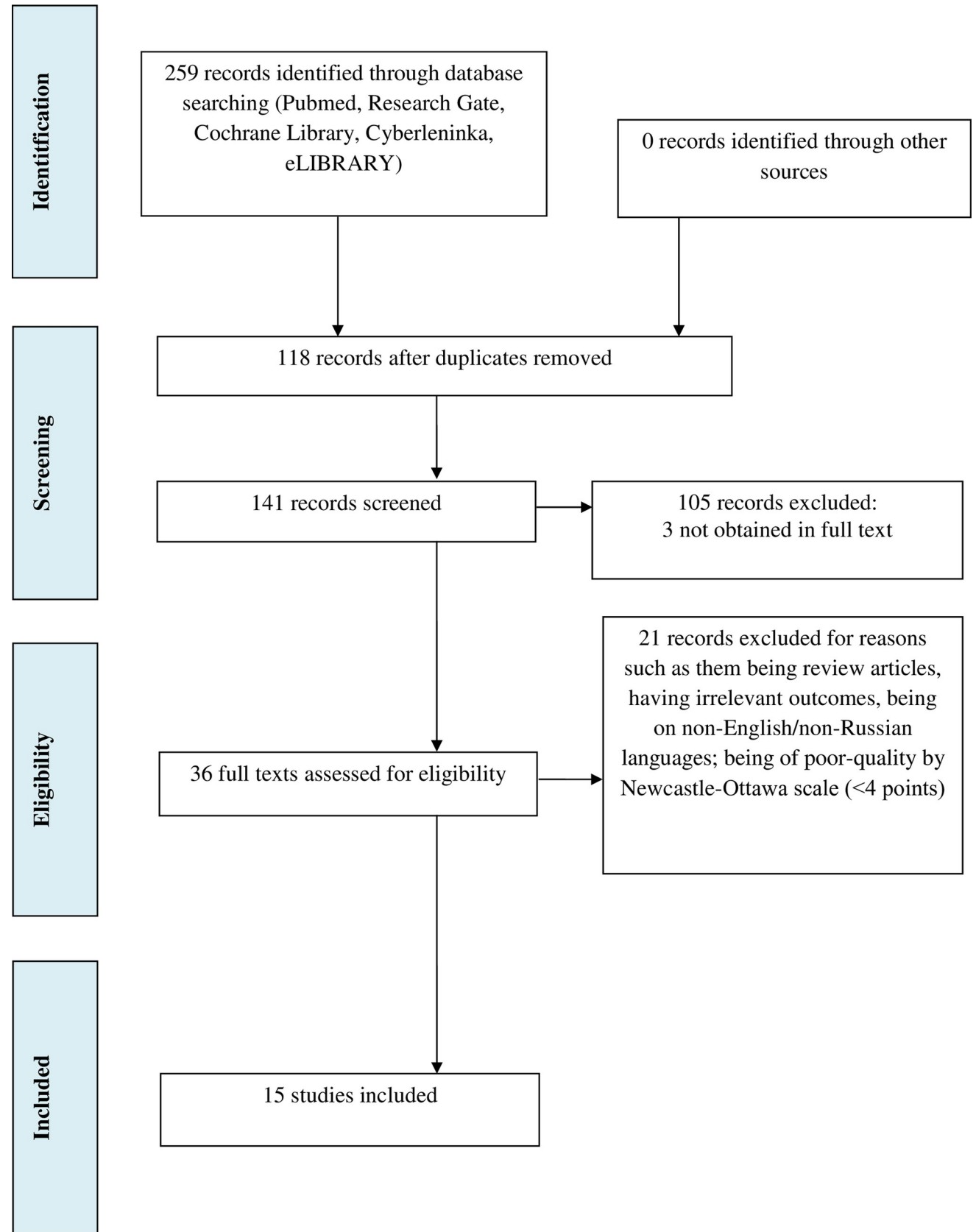

**Fig 1. Flow diagram of Preferred Reporting Items for Systematic Reviews and Meta-Analyses (PRISMA) presenting the process of search and selection of studies on prevalence of anxiety among vitiligo patients.**

groups, if available; (iv) prevalence of anxiety in vitiligo patients and in control groups, if available; (v) anxiety assessment tool; (vi) proportion of female and male patients with anxiety, if available. The quality of the included articles was assessed using the Newcastle-Ottawa scale for nonrandomized studies [15]. Studies that get $\geq 7$ points on Newcastle-Ottawa scale were considered to be of high quality, 4–6 points matched the criteria of medium-quality studies and <4 points were considered to be poor-quality studies. Each article was assessed independently by two authors (A.K. and L.K.). All scoring differences were conformed through discussion with Y.S.

## Statistical analysis

We carried out statistical analysis using the Review Manager software (RevMan) Version 5.3 (Copenhagen: The Nordic Cochrane Centre, The Cochrane Collaboration, 2014). Using the random-effects model, we then calculated the prevalence and event rate of anxiety in vitiligo patients and in total study populations. The $I^2$ value to assess heterogeneity among studies was processed. We used the random-effects model when heterogeneity across the studies was large ($I^2 > 60\%$, $P < 0.05$) and fixed-effects meta-analysis at small heterogeneity ($I^2 < 60\%$, $P < 0.05$).

We evaluated the risk of publication bias by using Egger's graphical method. Almost all studies (more than 95%) fall in symmetrical distribution that confirms minimal risks of selection bias. Thus, the funnel plot did not suggest any publication bias (see S1 Fig).

## Ethics statement

All analyses were based on previously published studies; no ethical approval or patient consent was required.

## Subgroup analyses

The subgroup analyses were performed to study the impact of gender, continent and skin disorder on the prevalence of anxiety among vitiligo patients. United Nations Standard Country or Area Codes for Statistical Use service was used to classify all included studies by continent [16]. We classified such countries as Saudi Arabia, United Arabian Emirates, Turkey and Georgia as countries in the Middle East. India and Bangladesh were grouped as Southern Asia countries. African region included Ethiopia, Nigeria and Egypt. Also, we differentiated Mexico as a Southern America country and Estonia as a European country. In controlled studies prevalence of anxiety was compared between vitiligo patients and non-vitiligo groups. As for the type of skin disorder, some studies contained the data on the prevalence of anxiety among patients with eczema, atopic dermatitis, psoriasis, albinism, melasma, acquired pigmentary disorders, acne, alopecia, chronic urticaria, neurodermatosis, scabies while the others contained the data on miscellaneous skin diseases. Most commonly the control group contained patients with psoriasis, acne and eczema (nine, six and four studies, respectively).

## Results

A search using Pubmed, Cochrane Library, Research Gate, Google Scholar, Cyberleninka and eLIBRARY allowed to identify 259 studies (Fig 1). Cochrane Library does not contain studies

which met the inclusion criteria. One hundred and eighteen studies were removed as duplicates.

The titles and abstracts of remaining 141 articles were screened. Of these, 105 were excluded from subsequent analysis as they did not provide enough data to calculate the effect size (102 publications) or could not be obtained in full text (3 publications). Full texts of 36 studies were reviewed, and 21 of them were excluded for the following reasons: irrelevant outcomes (n = 7); review articles (n = 7); non-English/non-Russian language (n = 4); poor-quality by Newcastle-Ottawa scale (<4 points) (n = 3). Finally, 15 articles with 1176 cases of vitiligo were included to our study and the sample size ranged from 15 to 164. All articles provided the data on the prevalence of anxiety in vitiligo patients. Anxiety was evaluated using self-report screening tools only.

The vast majority of studies (10) included participants whose age was >18 years, in two studies patients were elder than 15 years, in one study patients were elder than 16 years, one study comprised patients elder then 17 years, and in one study the age of participants was not mentioned. Six studies from Middle East, three studies from Southern Asia, four studies from Africa, one study from Southern America and one study from Europe were included. The number of female patients with vitiligo ranged from 23 to 103, in three studies the gender proportion was not stated. The tools which were used to identify the rate of anxiety were the Hospital Anxiety and Depression Scale (HADS), the fourth (DSM-IV) and the fifth (DSM-5) editions of Diagnostic and Statistical Manual of mental disorders, 28-item General Health Questionnaires, 21-item The Depression, Anxiety and Stress Scale (DASS-21), General Anxiety Disorder (GAD-7), ES-Q, Beck Depression Inventory, Illness Perception Questionnaire (IPQ) [4, 17–23].

All studies (n = 15) were designed as cross-sectional studies. Evaluating the quality of included studies with Newcastle-Ottawa scale, we found that three studies met the criteria of fair quality, twelve studies was of good quality. Study countries, geographical regions, sample size, prevalence of anxiety among vitiligo patients, screening tools, study design and quality of included publications are presented in Table 1.

Table 2 summarizes the prevalence and event rates of anxiety in total sample and in vitiligo patients. The sample size in fifteen studies varied from 42 to 618 participants. The number of vitiligo patients in included studies ranged from 23 to 164. The prevalence of anxiety in vitiligo patients fluctuated from 4.76% to 60.0%, so the minimal and maximal event rate meanings were equal to 0.05 and 0.60, respectively.

Analyses of the global rate of anxiety among vitiligo patients using random-effects models demonstrated that prevalence was equal to 35.8%. Also, we performed the subgroup analysis of anxiety rate among vitiligo patients according to their gender, type of skin disorder and continent of residence. The statistically significant difference in prevalence rates was found comparing African, European, Middle East and South Asian countries (33.29%, 27.93%, 32.02%, 13.73% respectively; P = 0.01 using $\chi2$- statistics) (see S2 Fig). However, there was no difference in prevalence rates based on patient gender and type of skin disorder in the subgroup analyses (see S3 Fig and S4 Fig).

The pooled odds ratio (OR) of anxiety among patients with vitiligo was 1.13 [95% CI 0.75, 1.70] (Fig 2). There was moderate heterogeneity between the studies ($I^2$ = 66%; $P$ = 0.0004).

The prevalence of anxiety was 40.38% for acne patients vs 32.93% for vitiligo patients, and this result was not statistically significant ($P$ = 0.22). The prevalence of anxiety in psoriasis patients was 27.34% vs vitiligo patients 28.44%. While that in eczema patients was 33.22% vs vitiligo patients 37.59%. However, these differences were not statistically significant (see S5 Fig).

There were only four studies that reported on prevalence rates of anxiety among female and male patients with vitiligo. The prevalence of anxiety in female patients compared to male

**Table 1. Location, prevalence of anxiety, screening tools, study design and quality of 18 included studies.**

| First Author, Year [ref] | Country | Continent | Sample Size | Number of vitiligo patients with anxiety | Prevalence of Anxiety,% | Instrument Used | Quality Rate | Study design |
|---|---|---|---|---|---|---|---|---|
| Abebe, 2016 [24] | Ethiopia | Africa | 154 | 60 | 38.8% | HADSa | 5 | Cross-sectional |
| Ahmed, 2016 [25] | Saudi Arabia | Middle East | 53 | 7 | 13.5% | DASS-21b | 7 | Cross-sectional |
| Ajose, 2014 [26] | Nigeria | Africa | 102 | 49 | 48% | HADSa | 6 | Cross-sectional |
| Al Ghamdi, 2010 [27] | Saudi Arabia | Middle East | 164 | 93 | 57% | IPQc | 7 | Cross-sectional |
| Alshahwan, 2015 [28] | Saudi Arabia | Middle East | 65 | 17 | 26,6% | HADSa | 5 | Cross-sectional |
| Ar Rashid, 2011 [29] | Bangladesh | Southern Asia | 50 | 6 | 12% | DSM-IVd | 5 | Cross-sectional |
| Balaban, 2011 [30] | Turkey | Middle East | 42 | 2 | 4.8% | HADSa | 6 | Cross-sectional |
| Dabas, 2019 [31] | India | Southern Asia | 95 | 19 | 21% | GAD-7e | 6 | Cross-sectional |
| Karelson, 2013 [32] | Estonia | Europe | 54 | 12 | 22% | ES-Qf | 5 | Cross-sectional |
| Karia, 2015 [33] | India | Southern Asia | 50 | 4 | 8% | GHQ 28g, DSM IVd | 5 | Cross-sectional |
| Morales-Sanchez, 2017 [11] | Mexico | Southern America | 150 | 90 | 60% | Beck Depression Inventory | 6 | Cross-sectional |
| Mufaddel, 2014 [34] | United Arabia Emirates | Middle East | 24 | 11 | 45.8% | HADS-Ah ICD-10 criteriai | 4 | Cross-sectional |
| Saleh, 2008 [35] | Egypt | Africa | 50 | 7 | 14% | GHQ 28g, Tailor Manifest Anxiety Scale, SDS questionnaire g | 7 | Cross-sectional |
| Sorour, 2017 [36] | Egypt | Africa | 108 | 34 | 31.48% | DSM-5k | 6 | Cross-sectional |
| Tsintsadze, 2015 [37] | Georgia | Middle East | 15 | 9 | 66.7% | HADSa | 5 | Cross-sectional |

Note.

a Hospital Anxiety and Depression Scale

b DASS-21, 21-item Depression Anxiety Stress Scales

c Illness Perception Questionnaire

d the fourth edition of Diagnostic and Statistical Manual of mental disorders

e Generalized Anxiety Disorder 7-item Scale

f ES-Q, Emotional State-Questionnaire

g 28-item General Health Questionnaire

h Hospital Anxiety and Depression Scale-Anxiety subscale

i ICD-10, International Statistical Classification of Diseases and Related Health Problems 10th Revision

g SDS, Zung Self-Rating Depression Scale

k DSM-5, the fifth edition of Diagnostic and Statistical Manual of mental disorders.

patients was higher (47.32% vs 42.4%) and this difference was statistically significant ($P = 0.03$) (see S6 Fig).

Grouping by continent the pooled OR among vitiligo and non-vitiligo patients was as follows: 1.82 [0.75, 4.40] in Africa ($P = 0.18$), 0.57 [0.25, 1.33] in Europe ($P = 0.19$), 0.80 [0.40, 1.63] in Middle East ($P = 0.54$), 1.32 [0.73, 2.40] in Southern Asia ($P = 0.36$) (see S7 Fig).

**Table 2. Prevalence and event rate of anxiety in total sample and in vitiligo patients.**

| Study, year | Number of vitiligo patients | Number of vitiligo patients with anxiety | Number of controls | Number of controls with anxiety | Sample size | Prevalence of anxiety in total sample | Prevalence of anxiety in vitiligo patients | Event rate in total sample | Event rate in vitiligo patients |
|---|---|---|---|---|---|---|---|---|---|
| Abebe, 2016 [24] | 154 | 60 | 464 | 150 | 618 | 33.98 | 38.96 | 0.34 | 0.39 |
| Ajose, 2014 [26] | 102 | 49 | 53 | 2 | 155 | 32.90 | 48.04 | 0.33 | 0.48 |
| ArRashid, 2011 [29] | 50 | 6 | 50 | 4 | 100 | 10.00 | 12.00 | 0.10 | 0.12 |
| Dabas, 2019 [31] | 95 | 20 | 0 | 0 | 95 | 21.05 | 21.05 | 0.21 | 0.21 |
| Karia, 2015 [33] | 50 | 4 | 50 | 2 | 100 | 6.00 | 8.00 | 0.06 | 0.08 |
| Karelson, 2013 [32] | 54 | 12 | 57 | 19 | 111 | 27.93 | 22.22 | 0.28 | 0.22 |
| Ahmed, 2016 [25] | 53 | 7 | 91 | 27 | 144 | 23.61 | 13.21 | 0.24 | 0.13 |
| AlGhamdi, 2010 [27] | 164 | 93 | 0 | 0 | 164 | 56.71 | 56.71 | 0.57 | 0.57 |
| Alshahwan, 2015 [28] | 65 | 17 | 459 | 132 | 524 | 28.44 | 26.15 | 0.28 | 0.26 |
| Balaban, 2011 [30] | 42 | 2 | 0 | 0 | 42 | 4.76 | 4.76 | 0.05 | 0.05 |
| Mufaddel, 2014 [34] | 24 | 11 | 55 | 14 | 79 | 31.65 | 45.83 | 0.32 | 0.46 |
| Saleh, 2008 [35] | 50 | 7 | 50 | 6 | 100 | 13.00 | 14.00 | 0.13 | 0.14 |
| Sorour, 2017 [36] | 108 | 34 | 506 | 187 | 614 | 35.99 | 31.48 | 0.36 | 0.31 |
| Tsintsadze, 2015 [37] | 15 | 9 | 103 | 81 | 118 | 76.27 | 60.00 | 0.76 | 0.60 |
| Morales-Sanchez, 2017 [11] | 150 | 90 | | 0 | 150 | 60.00 | 60.00 | 0.60 | 0.60 |

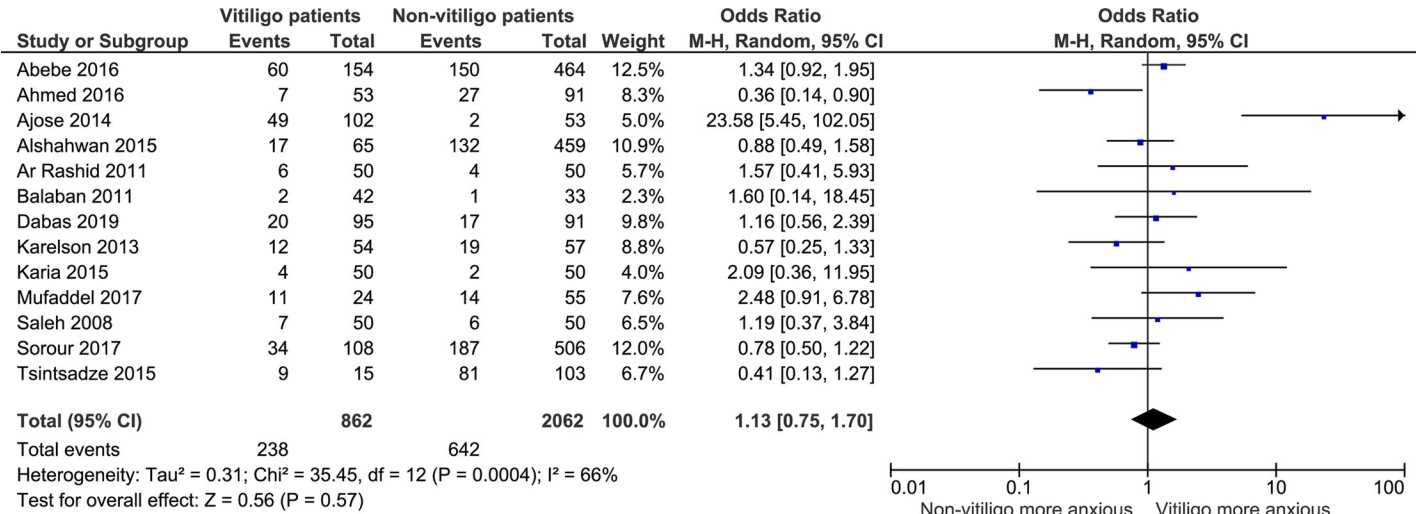

**Fig 2. Prevalence of anxiety compared between vitiligo patients and non-vitiligo groups (95% CI: = 95% Confidence Interval; M-H, Mantel-Haenszel Method).**

## Discussion

This meta-analysis aimed to evaluate the prevalence of anxiety among patients with vitiligo from different countries and to compare it with patients suffering from eczema, psoriasis, and acne. We expected that patients with vitiligo have a significantly higher risk of anxiety or anxiety symptoms as compared to those who present with no pigmentary skin disorders. In fact after conducted systematic review and meta-analysis the null hypothesis was reconfirmed, there was no difference in prevalence rates of anxiety among comparison groups. About one-third of vitiligo patients have the symptoms of anxiety. We found that the difference in the prevalence of anxiety between males and females was statistically significant, but of course the clinical relevance of this issue remains disputable. When comparing the prevalence of anxiety in different skin disorders including vitiligo, patients with psoriasis, eczema and acne had no the significantly higher rates of anxiety. Similarly, we did not find statistically significant difference in prevalence of anxiety between vitiligo patients coming from various continents.

Across the studies included in this meta-analysis, the prevalence of anxiety in vitiligo patients varied from 4.76% to 60.0%. Hospital Anxiety and Depression Score (HADS) was utilized in 40.0% of studies and it presents a self-assessment scale designed for detecting mental disorders in hospital settings [17]. The fourth (DSM-IV) and fifth (DSM-5) editions of Diagnostic and Statistical Manual of mental disorders were another popular method for anxiety detection in the selected articles. Other tools for anxiety measurement comprised 28-item version of General Health Questionnaires and this was used in two studies included in this meta-analysis. Also, the selected studies utilized other tools for anxiety detection, such as DASS-21, GAD-7, ES-Q, Beck Depression Scale, IPQ. The large variety of anxiety screening tools might be the possible cause of the moderate heterogeneity observed in our systematic review. Also, the wide range of anxiety assessment tools did not allow us to perform the subgroup analysis according to this criterion. Besides, due to the lack of data on prevalence of anxiety symptoms according to severity of vitiligo it was impossible to conduct the sub-analysis of anxiety rate regarding to extent of depigmentation.

Comparison of anxiety levels in vitiligo patients depending on the continent of their residence showed that there is no statistically significant difference in anxiety rate among people living in different regions. Although the highest odds ratio (1.82 [0.75, 4.40]) was found in African vitiligo patients. We suggest that this finding might be explained by various views and levels of acceptance typical for people belonging to different cultures. Due to the fact that ethnicity correlates with skin color, the presence of achromatic spots may have a greater adverse psychological effect on participants from ethnic groups with darker skin [38]. Certain cultures are characterized by a low level of vitiligo acceptance. For example, in some regions of India, vitiligo is often called "white leprosy", which leads to an even greater aggravation of the patient's psychological problem [7].

The difference in the prevalence of anxiety among male and female vitiligo patients was statistically significant as females had higher rate of anxiety. This fact is concordant with the global trends of anxiety epidemiology [39]. Borimnejad and co-authors found that female patients with vitiligo have lower quality of life indicators and more psychological disorders than male patients [40]. Similar results were also reported by the Tunisian study, where women with visible skin defects could not embody themselves as employee and potential bride due to stigmatization [41].

An analysis of the studies included in our meta-analysis demonstrated that anxiety is most often associated with psoriasis, eczema and acne as compared with other dermatological diagnoses. When comparing the prevalence of anxiety in patients with different skin disorders, we found that acne patients have the highest risk of anxiety development. Even though this

finding was not statistically significant a higher level of anxiety seen in acne patients could be explained by hormonal imbalance (elevated quantity of androgens) as well as by the fact that acne affects adolescents and young people at a time when they are more likely to be concerned about their body and social life [42]. Also, the prevalence of anxiety in psoriasis and vitiligo patients was very similar (27.34% vs. 28.44%, respectively). When comparing eczema and vitiligo patients, the prevalence rates of anxiety were almost comparable (33.22% vs.37.59%, respectively). Slightly higher prevalence of anxiety in vitiligo patients might be explained by the fact that most of the studies included in this meta-analysis were carried out in Asian and African countries and vitiligo in patients with dark skin phototype is associated with stigmatization and reduced quality of life [43].

Melanocytes destruction and anxiety seem to be associated strongly due to similar mechanisms of neuroendocrine dysregulation. Some studies demonstrate the role of increased levels of neuromediators in vitiligo development. Elevated levels of norepinephrine were found in microenvironment of melanocytes, urine and plasma of vitiligo patients [44]. These basic findings are confirmed by several descriptive studies. The higher levels of norepinephrine are detected in vitiligo patients at an active phase of disease [45]. Significant correlations between catecholamines' levels and progressive form of disease were described in 56 vitiliginous subjects [46]. The link between increased level of norepinephrine and destruction of melanocytes might be an essential element of oxidative stress theory of vitiligo pathogenesis. Mental stress causes activation of the hypothalamic–pituitary–adrenal axis, which secretes catecholamines. Monoamines and their metabolites stimulate α-receptors of skin arterioles, leading to microcirculation disturbances and hypoxia. As a result, overproduced oxygen radicals cause damage of melanocytes [47]. Also, the activity of acetylcholine esterase is lower in depigmented skin and grows during repigmentation process [44]. Schaullreuter explained a substantial contribution of $H_2O_2$-mediated oxidation of acetylcholine esterase to the oxidative stress in vitiligo [48]. Two studies were the most relevant to the topic of our study. Lai Y. et al. selected 25 studies with 2708 cases of vitiligo to find the prevalence of disease among them. Analysis of the results demonstrated that patients with vitiligo were significantly more likely to suffer from depressive disorders than healthy volunteers. The total prevalence of depression among patients with vitiligo was 0.253 (95% CI 0.16–0.34; $P<0.001$), and the pooled odds ratio was 5.05 against the control (95%CI 2.21–11.51; $P< 0,001$) [9]. The second meta-analysis was conducted by Osinubi and co-authors and was aimed at establishing the prevalence of psychological symptoms or disorders in people with vitiligo. This meta-analysis included 29 publications involving 2530 patients, and the authors found that approximately one in four people with vitiligo suffered from depression, and one in seven was affected by anxiety. The authors have repeatedly emphasized the heterogeneity of the studied groups and tools used and reported that the overall prevalence of anxiety ranged from 33% to 46% [12]. In fact, the lack of control over the screening tools utilized to measure anxiety is the major limitation of the previous meta-analyses and the cause of large heterogeneity. There are several differences between this meta-analysis and our work, which mostly relate to the primary aim. While we focused exclusively on anxiety, Osinubi and co-authors covered a broad range of psychological disorders: depression, anxiety, social phobia, agoraphobia. Second, we included more studies in our meta-analysis that is explained by a two-year interval between two studies during which a number of additional publications appeared.

Our study has several strengths and limitations. Firstly, we tried to minimize publication bias strictly following the rules of study selection for systematic reviews. Secondly, our meta-analysis includes studies with different screening tools from different countries and world regions. Thirdly, our study is dedicated to evaluation of the prevalence of anxiety exclusively among vitiligo patients. However, there are certain limitations and the main one is moderate

heterogeneity as a result of broad criteria for inclusion. This limitation derives from available publications on the study topic as they are often imprecise in identifying what is counted and not counted as anxiety in terms of type and severity. Due to this fact, the results of our study should be generalized with a caution. Secondly, the wide range of diagnostic tools may impact on study results. On our best knowledge, there is no unified and evidence-based tool for assessment of anxiety in patients with skin diseases. Finally, different cultural and social conditions in studies' environment may also contribute to heterogeneity of our meta-analysis. Nevertheless, it is well-known that psychological and psychiatric disorders might be strongly associated with ethnical, cultural and social factors [49, 50].

## Conclusions

In conclusion, the patients with vitiligo suffer from anxiety as frequently as do individuals with such severe skin conditions, as psoriasis or eczema. Although vitiligo patients present with no symptoms apart from decolorated skin patches, this pathology is accompanied with various psychological problems. Dermatologists and other specialists dealing with vitiligo patients should be aware of their predisposition to anxiety and be able to envisage correctional interventions to alleviate the burden of their mental distress. Clinical guidelines have to contain information about effective screening and management of anxiety among vitiligo patients. Finally, we did not find statistically significant impact of various risk factors on anxiety rate. This fact emphasizes the global burden of vitiligo that is not dependent on patient's gender or ethnicity.

## Supporting information

**S1 Checklist.**
(DOC)

**S1 Table. Search strategy Pubmed (Ovid MEDLINE).**
(TIF)

**S1 Fig. Funnel plots of studies evaluating the comparison: Anxiety in vitiligo and non-vitiligo patients.** SE, standard error (Log[OR]); OR, odds ratio.
(TIF)

**S2 Fig. Gender difference in anxiety prevalence in vitiligo patients (Al Ghamdi, 2010 [5], Dabas, 2019 [10]).**
(TIF)

**S3 Fig. Continent difference in anxiety prevalence in vitiligo patients.**
(TIF)

**S4 Fig. Disease difference in anxiety prevalence in vitiligo patients.**
(TIF)

**S5 Fig.** Meta-analysis of the prevalence of anxiety in patients with vitiligo compared with those with acne (a), psoriasis (b), eczema (c). (95% CI: = 95% Confidence Interval; M-H, Mantel-Haenszel Method.
(TIF)

**S6 Fig. Meta-analysis of the prevalence of anxiety in men and women.** (95% CI: = 95% Confidence Interval; M-H, Mantel-Haenszel Method).
(TIF)

**S7 Fig. Meta-analysis of the prevalence of anxiety depending on the continent.** (Chi2, Chi-Squared Test; df, degrees of freedom; Z, statistical test; I2, meta-analysis heterogeneity index; P < 0,05 (two-tailed)).
(TIF)

## Author Contributions

**Conceptualization:** Almira Akhmetova, Ulugbek Sabirov, Saltanat Adilgozhina, Raikhan Tuleutayeva, Yuliya Semenova.

**Data curation:** Almira Akhmetova, Ulugbek Sabirov, Saltanat Adilgozhina, Raikhan Tuleutayeva, Yuliya Semenova.

**Formal analysis:** Laura Kassym, Natalya Glushkova.

**Methodology:** Assiya Kussainova, Laura Kassym, Yuliya Semenova.

**Project administration:** Assiya Kussainova, Almira Akhmetova, Ulugbek Sabirov, Saltanat Adilgozhina, Raikhan Tuleutayeva, Yuliya Semenova.

**Resources:** Assiya Kussainova, Laura Kassym, Almira Akhmetova, Ulugbek Sabirov, Saltanat Adilgozhina.

**Software:** Laura Kassym, Natalya Glushkova.

**Supervision:** Almira Akhmetova, Natalya Glushkova, Saltanat Adilgozhina, Raikhan Tuleutayeva, Yuliya Semenova.

**Validation:** Assiya Kussainova, Yuliya Semenova.

**Visualization:** Assiya Kussainova, Laura Kassym, Natalya Glushkova.

**Writing – original draft:** Assiya Kussainova, Laura Kassym, Natalya Glushkova, Yuliya Semenova.

**Writing – review & editing:** Assiya Kussainova, Laura Kassym, Almira Akhmetova, Natalya Glushkova, Ulugbek Sabirov, Saltanat Adilgozhina, Raikhan Tuleutayeva, Yuliya Semenova.

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
