## [Decision Letter · Decision Letter 0]

23 Jul 2020

PONE-D-20-18423

Vitiligo and Anxiety: A Systematic Review and Meta-Analysis

PLOS ONE

Dear Dr. Kassym,

Thank you for submitting your manuscript to PLOS ONE. After careful consideration, we feel that it has merit but does not fully meet PLOS ONE’s publication criteria as it currently stands. Therefore, we invite you to submit a revised version of the manuscript that addresses the points raised during the review process.

Having intensively reviewed your draft, our external referees have indicated major drawbacks. Moreover, their final recommendations strongly differed, and, thus I have invited a further external referee and have double checked your submitted draft (see R #1, and comments given below), to come to a more balanced first decision. All in all, the indicated shortcomings are considered reasonable with regard to both PLOS ONE's quality standards and our readership's expectations. Therefore, we invite you to submit a revised version of the manuscript that addresses each and every point raised during the review process. Please note that a non-convincing revision (not considered acceptable with regard to language, reviewers' constructive criticism, content, generalizable outcome, and/or Authors' Guidelines) will lead to outright reject.

We look forward to receiving your revised manuscript.

Kind regards,

Andrej M Kielbassa

Academic Editor

PLOS ONE

Additional Editor Comments:

While some scientists occasionally have considered the systematic review (SR) as the top level of evidence for clinical science research in the past, please note that SR per se cannot be endorsed solely due to their format ("because they are a systematic review"), but must be critically evaluated for the problem reviewed in the respective article. Consequently, different qualities of RCTs (as given with your study, please see below), explicit statements of protocols, careful estimates of risk bias, large numbers of studies, and sophisticated statistical meta-analyses are not considered true substitutes for a thorough understanding of the literature, for critical analysis, and for scientifically sound and clinically relevant conclusions considered generalizable.The authors might additionally wish to go to R. Solow, Systematic review versus structured critical analysis, Cranio 1-13 (2019). https://doi.org/10.1080/08869634.2019.1614288, and discuss these problems.

Subgroup analyses would seem necessary, as well as analysis of heterogeneity. Please note that the latter is being consistently underestimated in meta-analyses, and one measure of heterogeneity is I², thus indicating the percentage of variance in your meta-analysis.

A major concern would be the quality of included papers. Please remember that the quality of a randomized trial, or, as it has been shown in monthly practice of publishing, the lack thereof, is not dependent on who is evaluating this trial. It is the trial is the trial is the trial, and, again, it is - the trial. In other words, some scientists would say: "If you put garbage in, there will be garbage out." With this in mind, the reviewer does not agree to "finally rely on studies with quality scores of 50% and above that will be included in this study".

Please note that there is a public trust that research will be conducted to the highest standards possible. The main drawback of the current submission, would seem that the standards discussed above might have been disregarded. Significant problems have been found with previous systematic reviews, thus questioning the reliability of the latter as the premier level of evidence for clinical science research. Just being aware of those deficiencies and drawbacks does not help - correct interpretation of valid clinical research should be done by the scientist, and not by the reader alone (even if the latter must always be alert). All other scenarios will lead wasting resources and time.

2. At this time, we ask that you please provide the full search strategy and search terms for at least one database used as Supplementary Information.

3. Please include the methodology and Results of any analyses conducted for the assessment of publication bias by Begg's funnel plot or and/or Egger's regression plot in your meta-analysis.

4. Please amend your list of authors on the manuscript to ensure that each author is correctly linked to each affiliation. Authors’ affiliations should reflect the institution where the work was done (if authors moved subsequently, you can also list the new affiliation stating “current affiliation:….” as necessary).

Reviewers' comments:

Reviewer's Responses to Questions

**Comments to the Author**

1. Is the manuscript technically sound, and do the data support the conclusions?

Reviewer #1: No

Reviewer #2: Partly

Reviewer #3: Yes

2. Has the statistical analysis been performed appropriately and rigorously? 

Reviewer #1: Yes

Reviewer #2: No

Reviewer #3: Yes

3. Have the authors made all data underlying the findings in their manuscript fully available?

Reviewer #1: No

Reviewer #2: No

Reviewer #3: Yes

4. Is the manuscript presented in an intelligible fashion and written in standard English?

Reviewer #1: Yes

Reviewer #2: No

Reviewer #3: Yes

5. Review Comments to the Author

Reviewer #1: Abstract

- "Statistically significant difference in anxiety rates was found among female and male patients (34.46% vs 32.78%) ( P =0.04)." Do the authors think that this difference would be clinically relevant?

- "The pooled odds ratio among vitiligo and non-vitiligo patients indicated a statistical significance in African countries (1.39 [1.07, 1.79]; P =0.01)." Do the authors think that this would be astonishing?

Intro

- "Unlike depression, anxiety among patients with vitiligo is less studied in the current literature." This alone would not justify any research on this topic.

- Althors have failed to convincingly elaborate both aims and objectives.

- A clear and indisputable null hypothesis is missing. remember that H0 must be deducible from the foregoing thoughts.

Results

- Please compare "Unlike depression, anxiety among patients with vitiligo is less studied in the current literature." with "Finally, 18 articles with 1419 cases of vitiligo were included to our study." Do you really think that this is a "less studied" context? This reviewer would not agree.

- "Thus, our meta-analysis includes 18 studies comprising 1419 vitiligo patients (...)." See above. Why do you repeat this information?

- Including cross sectional studies would seem difficult with respect to a valid outcome.

- "Evaluating the quality of included studies with Newcastle-Ottawa scale, we found that eight studies met the criteria of fair quality, the same number of studies was found to be of good quality, and two studies were of poor quality." This is considered too less quality for a valid conclusion. Please note that some scientists occasionally have considered the systematic review (SR) as the top level of evidence for clinical science research in the past. However, you should remember that SR per se cannot be endorsed solely due to their format ("because they are a systematic review"), and must be critically evaluated for the problem reviewed in the respective article. Consequently, different qualities of RCTs, explicit statements of protocols, careful estimates of risk bias, large numbers of studies, and sophisticated statistical meta-analyses are not considered true substitutes for a thorough understanding of the literature, for critical analysis, and for scientifically sound and clinically relevant conclusions.

Disc

- "We demonstrated that patients with vitiligo do not have a significantly higher risk of anxiety or anxiety symptoms as compared to those who present with no pigmentary skin disorders." Again, this would seem confirmative only, and is considered expectable. Thus, let me repeat my question: What rationale triggered your study?

- "This limitation derives from available publications on the study topic as theyare often imprecise in identifying what is (...)." This reviewer would like to agree. Some researchers would say "if you put garbage in, you will get garbage out". In other words, if the quality of included studies, is low, there will be a poor output only. However, in this case, doing such a research would seem doubtful.

Concl

- Please do not simply repeat your results here. Instead, provide a reasonable extension of your outcome.

Refs

- Please check Guidelines for Authors, and check uniform formatting.

Reviewer #2: Dear authors,

I appreciate your work, but the following data and analyzes are missing, and the following questions are open.

1. For all vitiligo patients

i2 in meta-analysis

2. For subgroups

Continents (prevalence and p-value for each continent)

Gender (prevalence and p-value for females and males)

Skin disorders (prevalence and p-value for the skin disorders you mentioned)

3. We used the random-effects model when heterogeneity across the studies was

112 large (i2> 60%, P< 0.05) and fixed-effects meta-analysis at small heterogeneity (i

2< 60%, P< 113 0.05).

Did you have a reference for this statement?

Is it not right to use random-effects model in your study?

4. How do you definite the event rate? What is your aim to report the event rate? The formula?

Is it not more logical to report the rate in a cross-sectional study?

Reviewer #3: It's a good quality systematic review. The main limitation of this study was the different screening tools used in the primary studies to identify anxiety symptoms. So I recommend to explain in the manuscript if you tried to do a subgroup analysis according to he anxiety scales apart from the meta-analysis for gender, control group and geographic regions. I also suggest to mention if you tried to do a meta-analysis acoording to the extent of vitiligo and the prevalence of anxiety symptoms.

6. PLOS authors have the option to publish the peer review history of their article (what does this mean?). If published, this will include your full peer review and any attached files.

Reviewer #1: No

Reviewer #2: **Yes: **Nasrin Hamidizadeh, PhD

Molecular Dermatology Research Center

Shiraz University of Medical Sciences

P.O. Box: 71348-44119

Shiraz, Iran

Tel: +98 71 32125592

Fax: +98 71 32319049

Email: n.hamidizadeh.mdrc@gmail.com

nhamidizadeh@gmx.net

Reviewer #3: **Yes: **Martha Alejandra Morales-Sánchez

---

## [Author Response · Author response to Decision Letter 0]

16 Aug 2020

Response to Reviewers:

Journal: PLoS ONE

Manuscript No: 

Manuscript title: Vitiligo and Anxiety: A Systematic Review and Meta-Analysis

Authors: Kussainova et al.

 Additional Editor Comments:

While some scientists occasionally have considered the systematic review (SR) as the top level of evidence for clinical science research in the past, please note that SR per se cannot be endorsed solely due to their format ("because they are a systematic review"), but must be critically evaluated for the problem reviewed in the respective article. Consequently, different qualities of RCTs (as given with your study, please see below), explicit statements of protocols, careful estimates of risk bias, large numbers of studies, and sophisticated statistical meta-analyses are not considered true substitutes for a thorough understanding of the literature, for critical analysis, and for scientifically sound and clinically relevant conclusions considered generalizable. The authors might additionally wish to go to R. Solow, Systematic review versus structured critical analysis, Cranio 1-13 (2019). https://doi.org/10.1080/08869634.2019.1614288, and discuss these problems. 

Thank you very much for considering our manuscript for publication. We treated all comments with all due respect and did our best to respond to the reviewers comments.

Subgroup analyses would seem necessary, as well as analysis of heterogeneity. Please note that the latter is being consistently underestimated in meta-analyses, and one measure of heterogeneity is I², thus indicating the percentage of variance in your meta-analysis. 

Thank you. We introduced the heterogeneity analysis.

A major concern would be the quality of included papers. Please remember that the quality of a randomized trial, or, as it has been shown in monthly practice of publishing, the lack thereof, is not dependent on who is evaluating this trial. It is the trial is the trial is the trial, and, again, it is - the trial. In other words, some scientists would say: "If you put garbage in, there will be garbage out." With this in mind, the reviewer does not agree to "finally rely on studies with quality scores of 50% and above that will be included in this study". 

We strongly agree with this statement and thus, excluded three poor quality studies from our meta-analysis.

Please note that there is a public trust that research will be conducted to the highest standards possible. The main drawback of the current submission, would seem that the standards discussed above might have been disregarded. Significant problems have been found with previous systematic reviews, thus questioning the reliability of the latter as the premier level of evidence for clinical science research. Just being aware of those deficiencies and drawbacks does not help - correct interpretation of valid clinical research should be done by the scientist, and not by the reader alone (even if the latter must always be alert). All other scenarios will lead wasting resources.

Thank you. Done. Please, see the changes listed below to evaluate the work done in this regard.

We are grateful for your personnel recommendations. It helped us to improve quality of our manuscript. Also revision process allowed us to find some technical mistakes which are successfully corrected (e.g. reference list in Manuscript). 

Thank you for considering our manuscript. We have addressed all the proposed amendments.

2. At this time, we ask that you please provide the full search strategy and search terms for at least one database used as Supplementary Information. 

Thank you. Done. 

Search Strategy Pubmed (see S1 Table)

No. Searches Results

1 “Vitiligo” [MeSH] OR hypomelanosis 11903

2 “Vitiligo” [MeSH] OR hypopigmentation 11664

3 “Vitiligo” [MeSH] AND “anxiety” 33

4 “Vitiligo” [MeSH] OR hypomelanosis AND “anxiety” 43

5 “Vitiligo” [MeSH] OR hypomelanosis AND “anxi*” 47

6 “Vitiligo” [MeSH] OR hypopigmentation AND “anxi*” 46

7 “Vitiligo” [MeSH] OR hypomelanosis AND “anxi*” AND “prevalence” 7

8 “Vitiligo” [MeSH] OR hypopigmentation AND “anxi*” AND “prevalence” 7

9 “Vitiligo” [MeSH] OR hypomelanosis AND “anxi*” AND “epidemiol” 18

10 “Vitiligo” [MeSH] OR hypopigmentation AND “anxi*” AND “epidemiol” 18

Search strategy for other databases includes utilizing of key words with subsequent selection of published manuscripts based on their titles and abstracts. Unfortunately, the use of Bullean operators and additional filters was not possible in above mentioned data bases.

3. Please include the methodology and Results of any analyses conducted for the assessment of publication bias by Begg's funnel plot or and/or Egger's regression plot in your meta-analysis. We evaluated publication bias using the Egger’s graphical method (Egger, M.; Davey Smith, G.; Schneider, M.; Minder, C. Egger, M.; Davey Smith, G.; Schneider, M.; Minder, C. Bias in meta-analysis detected by a simple, graphical test. BMJ 1997, 315, 629–634.)

S2 Fig represents the funnel plot of comparison: Anxiety in vitiligo and non-vitiligo patients, outcome: Anxiety. Almost all studies (more than 95%) fall in symmetrical distribution that confirms minimal risks of selection bias. Thus, the funnel plot did not suggest any publication bias (see S2 Fig).

 S2 Fig. Funnel plots of studies evaluating the comparison: anxiety in vitiligo and non-vitiligo patients. SE, standard error (Log[OR]); OR, odds ratio.

4. Please amend your list of authors on the manuscript to ensure that each author is correctly linked to each affiliation. Authors’ affiliations should reflect the institution where the work was done (if authors moved subsequently, you can also list the new affiliation stating “current affiliation:….” as necessary). 

Thank you. Done.

Reviewer 1

Abstract

Thank you very much for reviewing our manuscript. Your comments helped us to improve it significantly.

We highlighted all changes made in yellow

- "Statistically significant difference in anxiety rates was found among female and male patients (34.46% vs 32.78%) 

(P =0.04)." Do the authors think that this difference would be clinically relevant?

We excluded three articles from our systematic review due to their poor-quality, so some statistical findings have also been changed. 

Statistically significant difference in anxiety rates was found among female and male patients (34.46% vs 32.78% 47.32% vs 42.4%) (P=0.04 P=0.03), but the clinical relevance of this issue remains arguable.

- "The pooled odds ratio among vitiligo and non-vitiligo patients indicated a statistical significance in African countries (1.39 [1.07, 1.79]; P =0.01)." Do the authors think that this would be astonishing? Thank you very much for reviewing our manuscript. Your comments helped us to improve it significantly.

Also, 

The pooled odds ratio among vitiligo and non-vitiligo patients did not indicate a statistical significance among different continents. 

Intro

- "Unlike depression, anxiety among patients with vitiligo is less studied in the current literature." This alone would not justify any research on this topic.

Thank you. You're right. This statement was replaced by the following:

Although the sufficient number of papers devoted to the issue of anxiety in vitiligo patients had been published before, there is no pooled evidence that is needed for comprehensive understanding of this problem.

- Althors have failed to convincingly elaborate both aims and objectives.

To overcome this drawback, we added this information:

The specific goals of the present systematic review and meta-analysis are:

• To determine the prevalence (or event rate) of anxiety in vitiligo patients in comparison with non-vitiligo patients.

• To investigate the impact of some variables such as gender, continent, type of skin disorder on anxiety rate among vitiligo and non-vitiligo patients. 

- A clear and indisputable null hypothesis is missing. remember that H0 must be deducible from the foregoing thoughts.

In this systematic review the null hypothesis is that there is no difference in prevalence of anxiety among vitiligo and non-vitiligo persons.

Results

- Please compare "Unlike depression, anxiety among patients with vitiligo is less studied in the current literature." with "Finally, 18 articles with 1419 cases of vitiligo were included to our study." Do you really think that this is a "less studied" context? This reviewer would not agree.

Thank you. You're right. We have removed this sentence from the text.

- "Thus, our meta-analysis includes 18 studies comprising 1419 vitiligo patients (...)." See above. Why do you repeat this information?

Thank you. You're right. We have removed this sentence from the text.

- Including cross sectional studies would seem difficult with respect to a valid outcome.

As the aim of our systematic review was to identify any differences in anxiety rate among vitiligo and non vitiligo patients, the cross sectional studies were considered by us as relevant (or suitable) enough to extract the needed epidemiological data, in particular, the prevalence or event rate. 

- "Evaluating the quality of included studies with Newcastle-Ottawa scale, we found that eight studies met the criteria of fair quality, the same number of studies was found to be of good quality, and two studies were of poor quality." This is considered too less quality for a valid conclusion. Please note that some scientists occasionally have considered the systematic review (SR) as the top level of evidence for clinical science research in the past. However, you should remember that SR per se cannot be endorsed solely due to their format ("because they are a systematic review"), and must be critically evaluated for the problem reviewed in the respective article. Consequently, different qualities of RCTs, explicit statements of protocols, careful estimates of risk bias, large numbers of studies, and sophisticated statistical meta-analyses are not considered true substitutes for a thorough understanding of the literature, for critical analysis, and for scientifically sound and clinically relevant conclusions.

Thank you. You're right. Based on your recommendation, we excluded studies that were rated as "poor" on the Newcastle-Ottawa scale. This recommendation improved significantly the quality of our research. We reflected this in exclusion criteria statement:

Exclusion Criteria

Our exclusion criteria were as follows: (i) studies that did not state the rate of anxiety among vitiligo patients or provided insufficient data for calculation of anxiety rates; (ii) unavailability of the full text for full review; (iii) studies with low methodological quality, i.e. case reports, case series and commentaries; (iv) studies published in other languages apart from English and (v) studies that were considered to be of low-quality by Newcastle-Ottawa scale (˂4 points).

Disc

Thank you. We have added this information:

- "We demonstrated that patients with vitiligo do not have a significantly higher risk of anxiety or anxiety symptoms as compared to those who present with no pigmentary skin disorders." Again, this would seem confirmative only, and is considered expectable. Thus, let me repeat my question: What rationale triggered your study?

We expected that patients with vitiligo have a significantly higher risk of anxiety or anxiety symptoms as compared to those who present with no pigmentary skin disorders. In fact, after conducted systematic review and meta-analysis the null hypothesis rejecting any difference in prevalence rates of anxiety among comparison groups was reconfirmed.

- "This limitation derives from available publications on the study topic as they are often imprecise in identifying what is (...)." This reviewer would like to agree. Some researchers would say "if you put garbage in, you will get garbage out". In other words, if the quality of included studies, is low, there will be a poor output only. However, in this case, doing such a research would seem doubtful.

Thank you. You're right. Based on your recommendation, we excluded studies that were rated as "poor" on the Newcastle-Ottawa scale, as it was mentioned previously. 

Concl

Done. We totally revised this part of the manuscript:

- Please do not simply repeat your results here. Instead, provide a reasonable extension of your outcome.

In conclusion, the patients with vitiligo suffer from anxiety as frequently as do individuals with such severe skin conditions, as psoriasis or eczema. Although vitiligo patients present with no symptoms apart from decolorated skin patches, this pathology is accompanied with various psychological problems. Dermatologists and other specialists dealing with vitiligo patients should be aware of their predisposition to anxiety and be able to envisage correctional interventions to alleviate the burden of their mental distress. Clinical guidelines have to contain information about effective screening and management of anxiety among vitiligo patients. Finally, we did not find statistically significant impact of various risk factors on anxiety rate. This fact emphasizes the global burden of vitiligo that is not dependent on patient’s gender or ethnicity.

Refs

- Please check Guidelines for Authors, and check uniform formatting.

Thank you. Done.

Reviewer 2

I appreciate your work, but the following data and analyzes are missing, and the following questions are open.

Thank you for your kind comments. They helped us to improve the quality of our manuscript.

We highlighted all changes made in yellow

For all vitiligo patients i2 in meta-analysis

We excluded three articles from our systematic review due to their poor-quality, so some statistical findings have been changed. 

I2 = 66 % for all vitiligo patients i2 in meta-analysis

There was moderate heterogeneity between the studies (I2=62%; P= 0.0008) (I2=66%; P= 0.0004).

For subgroups

Continents (prevalence and p-value for each continent)

Gender (prevalence and p-value for females and males)

Skin disorders (prevalence and p-value for the skin disorders you mentioned)

We add additional information on subgroup analyses to the text.

Also, we performed the subgroup analysis of anxiety rate among vitiligo patients according to their gender, type of skin disorder and continent of residence. The statistically significant difference in prevalence rates was found comparing African, European, Middle East and South Asian countries (33.29%, 27.93%, 32.02%, 13.73% respectively; P= 0.01 using χ2- statistics) (see S3 Fig.). However, there was no difference in prevalence rates based on patient gender and type of skin disorder in the subgroup analyses (see S4 Fig. and S5 Fig.).

We used the random-effects model when heterogeneity 60% across the studies was

112 large (i2> 60%, P< 0.05) and fixed-effects meta-analysis at small heterogeneity (i2< 60%, P< 0.05).

Did you have a reference for this statement? 

I2 statistic is the test used to quantify heterogeneity and calculates the proportion of variation due to heterogeneity rather than due to chance. The I2 value ranges from 0% to 100%, with higher values indicating greater heterogeneity. As a rough guide, the I2 statistic can be interpreted as follows: 

• 0% to 40%: might not be important; 

• 30% to 60%: may represent moderate heterogeneity* 

• 50% to 90%: may represent substantial heterogeneity* • 75% to 100%: considerable heterogeneity* (https://cccrg.cochrane.org/sites/cccrg.cochrane.org/files/public/uploads/heterogeneity_subgroup_analyses_revising_december_1st_2016.pdf).

In our study, we assessed the range from 0 to 60% as a small heterogeneity, while other share as a large heterogeneity.

Is it not right to use random-effects model in your study?

We have used THE random-effects model for meta-analysis as this includes consideration of heterogeneity in the effect estimate (heterogeneity = i2 > 60%, P< 0.05).

A fixed-effect model assumes that there is no statistical heterogeneity between the studies. 

When studies are gathered from the published literature, the random-effects model is generally a more plausible match (https://www.meta-analysis.com/downloads/Meta-analysis%20Fixed-effect%20vs%20Random-effects%20models.pdf)

1. How do you definite the event rate? 

1. Event rate is the proportion of people with a specific condition in a group in whom the event is observed.

2. What is your aim to report the event rate? 

2. We aimed to report the event rate of anxiety to give comprehensive characteristics of total sample and the study group

The formula?

Formula:

a) Event rate of anxiety in total sample = total number of cases with anxiety/total number of sample

b) Event rate of anxiety in vitiligo patients = total number of vitiligo patients with anxiety/total of vitiligo patients

3. Is it not more logical to report the rate in a cross-sectional study?

3. Taken into account that this meta-analysis was objected to study epidemiological features of anxiety in vitiligo patients, we used both ‘Event rate’ and ‘Prevalence’ to characterize total population.

Reviewer 3

It's a good quality systematic review. The main limitation of this study was the different screening tools used in the primary studies to identify anxiety symptoms. 

Thank you very much for your comments, they really helped us to improve the quality of our manuscript.

We highlighted all changes made in yellow:

So I recommend to explain in the manuscript if you tried to do a subgroup analysis according to the anxiety scales apart from the meta-analysis for gender, control group and geographic regions.

We have added this information to the text of our manuscript:

Also, the wide range of anxiety assessment tools did not allow us to perform the subgroup analysis according to this criterion.

I also suggest to mention if you tried to do a meta-analysis acoording to the extent of vitiligo and the prevalence of anxiety symptoms.

Besides, due to the lack of data on prevalence of anxiety symptoms according to severity of vitiligo it was impossible to conduct the sub-analysis of anxiety rate regarding to extent of depigmentation.

---

## [Decision Letter · Decision Letter 1]

26 Aug 2020

PONE-D-20-18423R1

Vitiligo and Anxiety: A Systematic Review and Meta-Analysis

PLOS ONE

Dear Dr. Kassym,

thank you for re-submitting your revised manuscript to PLOS ONE. After careful consideration, we feel that it has improved, but does not fully meet PLOS ONE’s publication criteria as it currently stands. Therefore, we invite you to submit a re-revised version of the manuscript that addresses the points raised during the review process.

Having intensively reviewed your second draft, our external referees still have indicated minor and major drawbacks. All in all, the indicated shortcomings are considered reasonable with regard to both PLOS ONE's quality standards and our readership's expectations. Please note that a non-convincing re-revision (not considered acceptable with regard to language, reviewers' constructive criticism, content, generalizable outcome, and/or Authors' Guidelines) will lead to outright reject. Even if trying to be constructive, please remember that it is not considered the primary aim of our reviewers to act as co-authors of your manuscript.

We look forward to receiving your revised manuscript.

Kind regards,

Andrej M Kielbassa, Prof. Dr. med. dent. Dr. h. c.

Academic Editor

PLOS ONE

Reviewers' comments:

Reviewer's Responses to Questions

**Comments to the Author**

1. If the authors have adequately addressed your comments raised in a previous round of review and you feel that this manuscript is now acceptable for publication, you may indicate that here to bypass the “Comments to the Author” section, enter your conflict of interest statement in the “Confidential to Editor” section, and submit your "Accept" recommendation.

Reviewer #1: (No Response)

Reviewer #2: All comments have been addressed

Reviewer #3: All comments have been addressed

2. Is the manuscript technically sound, and do the data support the conclusions?

Reviewer #1: No

Reviewer #2: Yes

Reviewer #3: Yes

3. Has the statistical analysis been performed appropriately and rigorously? 

Reviewer #1: Yes

Reviewer #2: Yes

Reviewer #3: Yes

4. Have the authors made all data underlying the findings in their manuscript fully available?

Reviewer #1: Yes

Reviewer #2: Yes

Reviewer #3: Yes

5. Is the manuscript presented in an intelligible fashion and written in standard English?

Reviewer #1: No

Reviewer #2: Yes

Reviewer #3: Yes

6. Review Comments to the Author

Reviewer #1: This resubmitted draft has considerably improved with the first round of revisions. However, the current version still would not seem convincing, and is not considered ready to proceed.

Abstract

- Contents would seem confusing. See aims: "We aimed to evaluate the prevalence of anxiety among

6 patients with vitiligo and to elucidate the associated risk factors." Thus, anxiety and vitiligo was searched for. However, suddenly, other information will appear: "the anxiety rate for acne patients was higher than for vitiligo, eczema and psoriasis patients". Please revise thoroughly.

- "The pooled odds ratio among vitiligo and non-vitiligo patients did not indicate a statistical significance among different continents." Did you aim to find differences between continents. I guess you did not. Please revise.

- "In addition, the anxiety rate for acne patients was higher than for vitiligo, eczema and psoriasis patients, but it was not statistically significant." If was NOT statistically significant, would it be worth mentioning? Why would it be worth mentioning, if these differences would be by chance? Please revise.

- "This finding accentuates the necessity of anxiety awareness in management of vitiligo patients." There were no differences, right? And anxiety has no influence, right? So, why would "the necessity of anxiety awareness in management of vitiligo patients be accentuated"?

- In total, this section would not seem convincing. Please note that Abstract section and full text obviously do not correspond.

Intro

- "Globally, there is an increasing rate of anxiety disorders (...)." I would have some doubts on that postulated increase. Please compare to "The rate of anxiety disorder in a general population is around 12% (...)" in the same paragraph. This would seem contradicting.

- "In certain cultures individuals suffering from vitiligo may be stigmatized and could experience difficulty with finding a couple or staying employed." Reference missing. remember that each statement must be underlined with a reference.

- "Although the sufficient number (...)." What is "the" sufficient number?

- "Such, the existing data indicate that patients with vitiligo face a higher risk of mental distress, although the current evidence coming from pooled analyses is insufficient." How can you say that - if there is no evidence - patients with vitiligo would face a higher risk? This would not seem convincing.

- "A 2017 systematic review and meta-analysis on the prevalence of depression among patients with vitiligo estimated that the pooled prevalence of depression was 0.253 across 25 studies [9]. A 2018 meta-analysis of the prevalence and odds of depression in patients with vitiligo found a wide range of prevalence between 8% and 33% across 17 studies, depending on the diagnostic tool used. [10]." Please elaborate clearly what would be new with your study. From Table 1, it seems clear that you have included only one 2019 paper.

- Please revise for grammar and style. Must read "the null hypothesis was (...)."

- Again, please compare your Abstract section: The aims given in the Intro section do not correspond to your Results given in the Abstract section.

Meths

- Heterogeneity must read I².

- "(...) some studies contained the data on the prevalence of anxiety among patients with eczema, atopic dermatitis, (...)." But not all studies, right? However, how can you draw any conclusions from this? Again, please see Abstract section.

Results

- "The prevalence of anxiety was 40.38% for acne patients vs 32.93% for vitiligo patients (...)." Again, this would seem confusing, since your aims did not explain other (non-vitiligo) skin diseases. Please revise thoroughly.

Disc

- Again, please see comments given above: "This meta-analysis aimed at evaluation of the prevalence of anxiety among vitiligo patients (...)."

- "In fact after conducted systematic review and meta-analysis the null hypothesis rejecting any difference in prevalence rates of anxiety among comparison groups was reconfirmed." Meaning would seem unclear, please revise.

- "(...) we found that acne patients have the highest risk of anxiety development." This was NOT your aim, right?

- Same with psoriasis and eczema, right?

- Same with depression, right?

Refs

- Again, please revise for uniform formatting. Check tiles, and revise thoroughly.

In total, this revised draft would not seem ready to proceed.

Reviewer #2: Dear Authors,

1. I would suggest that in the manuscript, the authors uniformly use event rate or prevalence. Concerning the definition, the author provided for the event rate that is the same as prevalence; hence, it would be better to unify these terms in the whole manuscript.

2. Using a random effect is a correct solution in this manuscript, but the rationale incorrect. Using a fixed- or random-effect model based on the results of the heterogeneity test is a common mistake in the meta-analysis. The followings are the explanations regarding this important issue.

“Some researchers start the analysis by selecting the fixed-effect model. They then test-perform a statistical test for heterogeneity in effect sizes (the Q-test).

• If the test for heterogeneity is not statistically significant, they conclude that the fixed-effect model is consistent with the data, and use this model in the analysis.

• If the test for heterogeneity is statistically significant, they conclude that the fixed-effect model is not consistent with the data, and use the random-effects model in the analysis.

This approach is fundamentally flawed for two reasons.

Reason 1

If we want to choose a model based on the sampling frame, then we should select the model based on our understanding of how the studies sampled, and not the results of a statistical test. If we are working with studies that assess the impact of an intervention in different populations, then logic tells us that the random-effects model is the model that fits the data, and it’s the model that we should choose.

To suggest that a non-significant p-value justifies the use of a fixed-effect analysis is to recommend that the lack of significance proves that the null is correct (that the studies share a common effect size). As we all learned in our first statistics class, the lack of significance does not prove that the null is true. And here, logic tells us that the null is probably false.

Reason 2.

The “flawed” approach uses the fixed-effect model as the starting point and requires evidence (a significant test of heterogeneity) to shift to the random-effects model.

The random-effects model should be the logical starting point. The random-effects model says that the true effect size may or may not vary from study to study, and thus does not assume that either is the case. As part of the analysis, we estimate the amount of variance in true effects across studies, and the estimate may or may not be zero.

By contrast, the fixed-effect model requires that the true effect size does not vary from study to study. Therefore, the fixed-effect model is more restrictive. It imposes a constraint that is neither necessary nor plausible.

If we should be using the random-effects model and (by mistake) employ the fixed-effect model, then it’s likely that

• The estimate of the mean will be incorrect

• The standard error will be incorrect

• The test of significance for mean will be incorrect

• The confidence interval about the mean effect will be too narrow

More fundamentally, the choice of a model defines the goals of the analysis.

The choice of a model determines the meaning of the summary effect

• Under the fixed-effect model, there is only one true effect. The summary effect is an estimate of that value.

• Under the random-effects model, there is a distribution of true effects. The summary effect is an estimate of that distribution’s mean.

One of the most important goals of a meta-analysis is to determine how the effect size varies across studies.

• When we use the fixed-effect model, we can estimate the common effect size, but we cannot discuss how the effect size varies, since this model assumes that the true effect size is the same in all studies.

• By contrast, if we elect to work with the random-effects model, we can ask not only “What is the mean effect size” but also “How does the effect size vary across populations.” In many cases, this question is key to understanding the effectiveness of the intervention.”

Conclusion

In other words, the analyses are correct, but when there is the heterogeneity, is random effects applied.

Good luck

Reviewer #3: The authors answered my questions, good effort. However the main limitation of this study remains to be the way the included studies measured the presence of anxiety.

7. PLOS authors have the option to publish the peer review history of their article (what does this mean?). If published, this will include your full peer review and any attached files.

Reviewer #1: No

Reviewer #2: **Yes: **Hamidizadeh Nasrin

Reviewer #3: No

---

## [Author Response · Author response to Decision Letter 1]

6 Oct 2020

Reviewer 1

This resubmitted draft has considerably improved with the first round of revisions. However, the current version still would not seem convincing, and is not considered ready to proceed.

Thank you very much for re-reviewing our manuscript. You have done a great job and we will try to give detailed answers to all comments. We highlighted all changes made in yellow

Abstract

- Contents would seem confusing. See aims: "We aimed to evaluate the prevalence of anxiety among patients with vitiligo and to elucidate the associated risk factors." Thus, anxiety and vitiligo was searched for. However, suddenly, other information will appear: "the anxiety rate for acne patients was higher than for vitiligo, eczema and psoriasis patients". Please revise thoroughly.

Thank you. Done. We have paraphrased the aim of our study

We aimed to evaluate the prevalence of anxiety among patients with vitiligo from different countries and to compare it with patients suffering from eczema, psoriasis, and acne. and to elucidate the associated risk factors.

- "The pooled odds ratio among vitiligo and non-vitiligo patients did not indicate a statistical significance among different continents." Did you aim to find differences between continents. I guess you did not. Please revise.

Done. We made this statement more concrete:

In addition, the pooled odds ratio among vitiligo and non-vitiligo patients did not indicate a statistical significance among patients coming from different continents.

- "In addition, the anxiety rate for acne patients was higher than for vitiligo, eczema and psoriasis patients, but it was not statistically significant." If was NOT statistically significant, would it be worth mentioning? Why would it be worth mentioning, if these differences would be by chance? Please revise.

Thank you. We have removed this sentence from the text.

- "This finding accentuates the necessity of anxiety awareness in management of vitiligo patients." There were no differences, right? And anxiety has no influence, right? So, why would "the necessity of anxiety awareness in management of vitiligo patients be accentuated"?

Thank you. We have paraphrased this part of the conclusion:

This finding accentuates the necessity of anxiety awareness in management of vitiligo patients with skin diseases.

- In total, this section would not seem convincing. Please note that Abstract section and full text obviously do not correspond.

Thank you. The abstract was revised according to your recommendations

Intro

- "Globally, there is an increasing rate of anxiety disorders (...)." I would have some doubts on that postulated increase. Please compare to "The rate of anxiety disorder in a general population is around 12% (...)" in the same paragraph. This would seem contradicting.

Thank you. We have removed this sentence from the text.

The rate of anxiety disorder in a general population is around 12% and certain categories of people are predisposed to it to a higher extent [5].

- "In certain cultures individuals suffering from vitiligo may be stigmatized and could experience difficulty with finding a couple or staying employed." Reference missing. remember that each statement must be underlined with a reference.

Thank you. We added the reference after this statement. 

39. Porter JR, Beuf AH, Lerner A, Nordlund J. Psychosocial effect of vitiligo: A comparison of vitiligo patients with “normal” control subjects, with psoriasis patients, and with patients with other pigmentary disorders. J Am Acad Dermatol. 1986;15(2):220–224. doi:10.1016/s0190-9622(86)70160-6.

- "Although the sufficient number (...)." What is "the" sufficient number?

Thank you. We have paraphrased this sentence.

Although the sufficient number relatively many papers devoted to the issue of anxiety in vitiligo patients had been published before, there is no pooled evidence that is needed for comprehensive understanding of this problem.

- "Such, the existing data indicate that patients with vitiligo face a higher risk of mental distress, although the current evidence coming from pooled analyses is insufficient." How can you say that - if there is no evidence - patients with vitiligo would face a higher risk? This would not seem convincing.

Thank you. We have paraphrased this sentence.

Such, the existing data indicate that patients with vitiligo possibly face a higher risk of mental distress, although the current evidence coming from pooled analyses is insufficient. In this systematic review the null hypothesis is that there is no difference in prevalence of anxiety among vitiligo and non-vitiligo persons.

- "A 2017 systematic review and meta-analysis on the prevalence of depression among patients with vitiligo estimated that the pooled prevalence of depression was 0.253 across 25 studies [9]. A 2018 meta-analysis of the prevalence and odds of depression in patients with vitiligo found a wide range of prevalence between 8% and 33% across 17 studies, depending on the diagnostic tool used. [10]." Please elaborate clearly what would be new with your study. From Table 1, it seems clear that you have included only one 2019 paper.

Firstly, in above mentioned studies the authors aimed to find the prevalence of depression in vitiligo patients, while in our study we aimed to find the prevalence of anxiety. 

Secondly, depression is not always associated with anxiety.

And finally, even one additional paper can impact significantly on the result of meta analysis. 

- Please revise for grammar and style. Must read "the null hypothesis was (...)."

Thank you. Done. 

In this systematic review the null hypothesis is was that there is no difference in prevalence of anxiety among vitiligo and non-vitiligo persons.

- Again, please compare your Abstract section: The aims given in the Intro section do not correspond to your Results given in the Abstract section.

Thank you. Done. 

Therefore, the aim of this study was to evaluate the prevalence of anxiety among patients with vitiligo from different countries and to compare it with patients suffering from eczema, psoriasis, and acne and to elucidate the associated risk factors by conducting a systematic review and meta-analysis of published observational studies. 

Meths

- Heterogeneity must read I².

Thank you. Done. 

The i2 I2 value to assess heterogeneity among studies was processed. We used the random-effects model when heterogeneity across the studies was large (i2 I2 > 60%, P< 0.05) and fixed-effects meta-analysis at small heterogeneity (i2 I2 < 60%, P< 0.05).

- "(...) some studies contained the data on the prevalence of anxiety among patients with eczema, atopic dermatitis, (...)." But not all studies, right? However, how can you draw any conclusions from this? Again, please see Abstract section.

Several studies involve patients with eczema, psoriasis, and acne as comparators for the vitiligo group. Subsequently, these skin diseases were included to subgroup analysis. Other studies contain the data on the prevalence of anxiety in other control groups (e.g. healthy controls), but the number of participants was not sufficient for a subgroup analysis.

Results

- "The prevalence of anxiety was 40.38% for acne patients vs 32.93% for vitiligo patients (...)." Again, this would seem confusing, since your aims did not explain other (non-vitiligo) skin diseases. Please revise thoroughly.

Thank you. We have paraphrased the aim of our study.

Disc

- Again, please see comments given above: "This meta-analysis aimed at evaluation of the prevalence of anxiety among vitiligo patients (...)."

Thank you. We have paraphrased the aim of our study.

This meta-analysis aimed to evaluate the prevalence of anxiety among patients with vitiligo from different countries and to compare it with patients suffering from eczema, psoriasis, and acne. and to elucidate the associated risk factors.

- "In fact after conducted systematic review and meta-analysis the null hypothesis rejecting any difference in prevalence rates of anxiety among comparison groups was reconfirmed." Meaning would seem unclear, please revise.

Thank you. We have revised this sentence.

"In fact after conducted systematic review and meta-analysis the null hypothesis was reconfirmed, there was no difference rejecting any difference in prevalence rates of anxiety among comparison groups was reconfirmed."

- "(...) we found that acne patients have the highest risk of anxiety development." This was NOT your aim, right?

We have paraphrased the aim of our study.

- Same with psoriasis and eczema, right?

We have paraphrased the aim of our study.

- Same with depression, right?

The aim of our study was to find the pooled prevalence of anxiety, but not the depression.

Refs

- Again, please revise for uniform formatting. Check tiles, and revise thoroughly.

Thank you. All references were drown up according to author instructions. 

In total, this revised draft would not seem ready to proceed.

Thank you. We have revised our draft according to your recommendations.

Reviewer 2

Dear Authors,

1. I would suggest that in the manuscript, the authors uniformly use event rate or prevalence. Concerning the definition, the author provided for the event rate that is the same as prevalence; hence, it would be better to unify these terms in the whole manuscript.

1. We have deleted event rate and describe only prevalence in the study.

To determine the prevalence (or event rate) of anxiety in vitiligo patients in comparison with non-vitiligo patients.

2. Using a random effect is a correct solution in this manuscript, but the rationale incorrect. Using a fixed- or random-effect model based on the results of the heterogeneity test is a common mistake in the meta-analysis. The followings are the explanations regarding this important issue.

“Some researchers start the analysis by selecting the fixed-effect model. They then test-perform a statistical test for heterogeneity in effect sizes (the Q-test).

• If the test for heterogeneity is not statistically significant, they conclude that the fixed-effect model is consistent with the data, and use this model in the analysis.

• If the test for heterogeneity is statistically significant, they conclude that the fixed-effect model is not consistent with the data, and use the random-effects model in the analysis.

This approach is fundamentally flawed for two reasons.

Reason 1

If we want to choose a model based on the sampling frame, then we should select the model based on our understanding of how the studies sampled, and not the results of a statistical test. If we are working with studies that assess the impact of an intervention in different populations, then logic tells us that the random-effects model is the model that fits the data, and it’s the model that we should choose.

To suggest that a non-significant p-value justifies the use of a fixed-effect analysis is to recommend that the lack of significance proves that the null is correct (that the studies share a common effect size). As we all learned in our first statistics class, the lack of significance does not prove that the null is true. And here, logic tells us that the null is probably false.

Reason 2.

The “flawed” approach uses the fixed-effect model as the starting point and requires evidence (a significant test of heterogeneity) to shift to the random-effects model.

The random-effects model should be the logical starting point. The random-effects model says that the true effect size may or may not vary from study to study, and thus does not assume that either is the case. As part of the analysis, we estimate the amount of variance in true effects across studies, and the estimate may or may not be zero.

By contrast, the fixed-effect model requires that the true effect size does not vary from study to study. Therefore, the fixed-effect model is more restrictive. It imposes a constraint that is neither necessary nor plausible.

If we should be using the random-effects model and (by mistake) employ the fixed-effect model, then it’s likely that

• The estimate of the mean will be incorrect

• The standard error will be incorrect

• The test of significance for mean will be incorrect

• The confidence interval about the mean effect will be too narrow

More fundamentally, the choice of a model defines the goals of the analysis.

The choice of a model determines the meaning of the summary effect

• Under the fixed-effect model, there is only one true effect. The summary effect is an estimate of that value.

• Under the random-effects model, there is a distribution of true effects. The summary effect is an estimate of that distribution’s mean.

One of the most important goals of a meta-analysis is to determine how the effect size varies across studies.

• When we use the fixed-effect model, we can estimate the common effect size, but we cannot discuss how the effect size varies, since this model assumes that the true effect size is the same in all studies.

• By contrast, if we elect to work with the random-effects model, we can ask not only “What is the mean effect size” but also “How does the effect size vary across populations.” In many cases, this question is key to understanding the effectiveness of the intervention.”

Conclusion

In other words, the analyses are correct, but when there is the heterogeneity, is random effects applied.

Good luck

2. We assume that the studies included in our meta-analysis using random-effects model could be a random sample from the set of studies. Also, we assume that the results of each study included in the meta-analysis represent the effect size for a particular study, which may vary within the population mean effect size. [Cohn LD, Becker BJ. How meta-analysis increases statistical power. Psychol Methods. 2003;8:243-253. http://dx.doi. org/10.1037/1082-989X.8.3.243; Fleiss JL. The statistical basis of meta-analysis. Stat Methods Med Res. 1993;2:121-145.].

We can suggest that the results of each study in this meta-analysis represent a unique effect. Because of this assumption, larger studies are given proportionally less weight, and, vice versa the smaller studies are given proportionally more weight [Borenstein M, Hedges L, Higgins J, Rothstein H. Introduction to Meta-Analysis. Chichester, West Sussex, UK: John Wiley & Sons; 2009]. Thus, through the use of a random effects model, the unique effect of each study was considered for the calculations. [Deeks JJ, Altman D, Bradburn M. Statistical methods for examining heterogeneity and combining results from several studies in metaanalysis. In: Egger M, Smith GD, Altman D, eds. Systematic Reviews in Health Care: MetaAnalysis in Context. London, UK: BMJ Books; 2001:285-312; Fleiss JL. The statistical basis of meta-analysis. Stat Methods Med Res. 1993;2:121-145]. 

Since, when testing the included studies data, we found statistical and epidemiological heterogeneity, therefore we used the random effects model. This allows to weigh more evenly the studies included in the meta-analysis. As a result smaller studies will have a greater relative impact on the combined total effect than fixed effect model [Sterne JA, Egger M, Moher D. Addressing reporting biases. In: Higgins JP, Green S, eds. Cochrane Handbook for Systematic Reviews of Interventions. Chichester, West Sussex, UK: John Wiley & Sons; 2008:297-333.]. 

Thus, we are able to assess the effect of anxiety in population, taking into account the fact that it can differ significantly between groups and represent a study-specific effect size.

Reviewer 3

Thank you very much for your comments.

Unfortunately, this problem is common for the vast majority of systematic reviews and meta-analyses. However, we tried to consider all possible factors underlying the heterogeneity of studies included in our review using random-effects and fixing-effects models.

---

## [Decision Letter · Decision Letter 2]

15 Oct 2020

Vitiligo and Anxiety: A Systematic Review and Meta-Analysis

PONE-D-20-18423R2

Dear Dr. Kassym,

We’re pleased to inform you that your manuscript has been judged scientifically suitable for publication and will be formally accepted for publication once it meets all outstanding technical requirements.

Once again, thank you very much for submitting this valuable paper to PLoS One. Please note that your draft will be forwarded to the publisher now, and you will receive the respective page proofs (these might be re-edited with regard to some minor editing improvements, so please be prepared to double check these proofs very carefully) in due course. 

Congratulations, best regards, and stay healthy, please!

Andrej M Kielbassa, Prof. Dr. med. dent. Dr. h. c.

Academic Editor

PLoS One

Reviewers' comments:

Reviewer's Responses to Questions

**Comments to the Author**

1. If the authors have adequately addressed your comments raised in a previous round of review and you feel that this manuscript is now acceptable for publication, you may indicate that here to bypass the “Comments to the Author” section, enter your conflict of interest statement in the “Confidential to Editor” section, and submit your "Accept" recommendation.

Reviewer #1: All comments have been addressed

Reviewer #2: All comments have been addressed

Reviewer #3: All comments have been addressed

2. Is the manuscript technically sound, and do the data support the conclusions?

Reviewer #1: Yes

Reviewer #2: Yes

Reviewer #3: Yes

3. Has the statistical analysis been performed appropriately and rigorously? 

Reviewer #1: Yes

Reviewer #2: Yes

Reviewer #3: Yes

4. Have the authors made all data underlying the findings in their manuscript fully available?

Reviewer #1: Yes

Reviewer #2: Yes

Reviewer #3: Yes

5. Is the manuscript presented in an intelligible fashion and written in standard English?

Reviewer #1: Yes

Reviewer #2: No

Reviewer #3: Yes

6. Review Comments to the Author

Reviewer #1: The authors have thoroughly addressed all reviewers' comments, and have improved their draft considerably.

Reviewer #2: The authors would like to choose either event rate or prevalence.

A minor edit in English is also recommended.

Reviewer #3: Your manuscript really improved from the first review and include the main limitations of your systematic review and metaanalysis.

7. PLOS authors have the option to publish the peer review history of their article (what does this mean?). If published, this will include your full peer review and any attached files.

Reviewer #1: No

Reviewer #2: **Yes: **Hamidizadeh Nasrin

Reviewer #3: No

---

## [Editor Report · Acceptance letter]

19 Oct 2020

PONE-D-20-18423R2 

Vitiligo and Anxiety: A Systematic Review and Meta-Analysis 

Dear Dr. Kassym:

I'm pleased to inform you that your manuscript has been deemed suitable for publication in PLOS ONE. Congratulations! Your manuscript is now with our production department. 

Kind regards, 

on behalf of

Prof. Dr. med. dent. Dr. h. c. Andrej M Kielbassa 

Academic Editor

PLOS ONE